# SyllableLM: Learning Coarse Semantic Units for Speech Language Models

## Abstract

Self-Supervised Transformer Models are the backbone of much of the recent progress in deep learning. However, these models require their inputs to be tokenized, and tokenization strategies for continuous data like audio and vision are often based on simple heuristics such as fixed sized convolutions or discrete clustering. For speech and audio models in particular, the high resolution of waveforms (16,000 samples/second or more) presents a significant challenge, as several times more tokens are used per word than in textual language modeling. In this work, we introduce a controllable, fully-self-supervised technique to dynamically merge speech representations across time to as low as 5 Hz at 60 bits per second while still preserving semantic information. We do this by 1) extracting noisy boundaries through analyzing correlations between mask spans and model losses and 2) iteratively improving these representations with a novel agglomeration technique. Using these new feature representations, we successfully train SyllableLM, a Neural Codec Language Model (NCLM) competitive with current SoTA NCLMs on a range of common benchmarks with a 30x reduction in pretraining compute, 5x reduction in inference compute, and 2.5x reduction in bitrate.

## 1 Introduction

Self-Supervised Learning (SSL) seeks to learn powerful, abstract representations of data without external labels. These representations can then be used in downstream tasks to achieve high performance even when modest amounts of supervised fine-tuning data are available. In audio and speech processing, a key motivation for this learning paradigm is the fact that young children learn to listen and speak well before they can read or write. While current textual language models [52, 59, 9] can compose highly realistic text, the research community has not yet developed similarly performant models that learn solely from spoken language. An increasing focus has coalesced around Generative Spoken Language Modeling (GSLM) [34], which sets out to achieve this goal.

The most successful of these approaches are autoregressive decoder transformer models [53] such as AudioLM [8] and TWIST [26], which operate on tokens learned through quantizing the output of SSL encoder models [28, 14]. However, these self-supervised tokenizations are much denser than their textual counterparts with the token rates typically between 25 and 50 tokens per second for speech models, as opposed to the typical human speaking rate of 2-5 words per second. The long context lengths that result from high temporal resolution tokenizations in speech models substantially impair both pretraining and inference speed, and it is additionally unclear to what extent modeling speech with a high granularity harms more abstract semantic understanding.

Very recently, there has been significant progress in extracting coarser speech unit representations from raw audio. In particular, SD-HuBERT [12] distills HuBERT [28] using only audio with a DINO-like distillation objective, and VG-HuBERT [45, 46] uses a contrastive loss against cross-modal

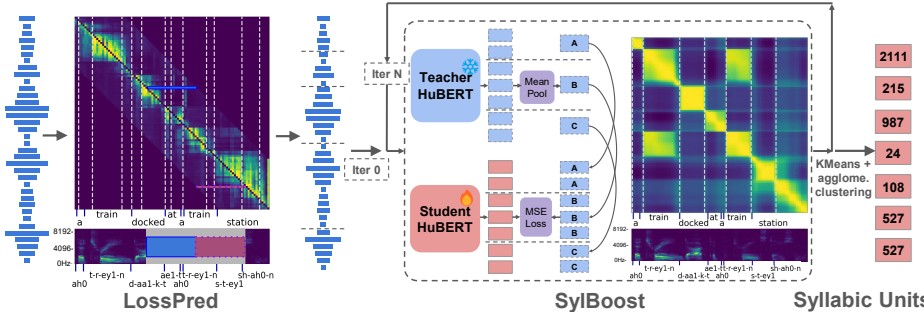

Figure 1: Left-Top: The loss prediction matrix $C$, where brighter is higher likelihood placed on the teacher label. A time-aligned transcript is on the bottom, and predicted cluster unit boundaries span vertically as dashed-lines. Left-Bottom: A Mel-Spectrogram of the input waveform with an example masked timespan in gray. The losses on tokens at timesteps covered by the solid blue and dotted red spans are mapped to their corresponding rows and columns in $C$ as described in Section 3.1. Right: Visual of our agglomeration procedure. We train a student to match intermediate teacher features pooled over regions generated by pseudo-syllable-boundaries. We use a min-cut algorithm to extract boundaries, and then apply K-Means and Agglomerative clustering to obtain discrete units.

visual inputs. We continue and significantly improve upon this line of research, resulting in the first syllable-like units suitable for high-quality GSLM. Specifically, we demonstrate breakthrough improvements in textual reconstruction from low-bitrate units of SSL models, reducing the word-error-rate (WER) from 37% using SD-HuBERT units to 7%, and more than halving realized bitrate of previous SpeechLM units from 175Bps to as low as 60Bps. We additionally find that our units correlate strongly with syllables both in boundary detection and in cluster quality.

Furthermore, we evaluate the effects of training SpeechLMs on these new units and obtain state-of-the-art results across a wide-variety of metrics, competitive with or outperforming AudioLM (350M parameters) and all TWIST model sizes (125M-13B parameters) with fewer parameters and fewer GPU-Hours. We commit to making our code open-source and plan to release our tokenizer and SpeechLM parameters. Our contributions are as follows:

1. We propose a novel training-free algorithm named LossPred that reveals noisy syllabic-like segmentation of unannotated speech signals by analyzing the loss of a pretrained self-supervised model (e.g. HuBERT) across different masking spans.
2. We propose a novel bootstrapping framework for speech unit quantization named SylBoost that achieves SotA unsupervised syllabic segmentation, categorization, and low-bitrate unit-to-audio resynthesis.
3. Using quantized SylBoost units as a basis for tokenization, we train SyllableLM, a generative spoken language model that outperforms or matches AudioLM and TWIST on a range of tasks while being 30x faster to train, 5x faster for inference, and having a 2.5x reduction in unit bitrate.

## 2 Related Work

**Self-Supervised Encoder Models**    There has been a great amount of work in learning high-level representations from data by reconstructing corrupted inputs across speech [3, 28, 6], audio [24], text [20, 15], and vision [10, 27]. To navigate the lack of simple discrete targets in speech, much work has been placed in finding high-quality targets, such as iterative clustering [28] and by predicting the feature representations of a teacher network based on a running average of student model weights [5, 6]. An alternate but similar line of work has been placed into learning low-bitrate units for the task of resynthesis [19, 58, 56, 33, 60, 21], which include losses focused on reconstruction and use an information bottleneck to enforce compression.

**Applications of Neural Codecs**    The discrete units generated by these self-supervised encoders are versatile and fundamental to much of the recent progress in speech research such as Text-To-Speech

[54, 29, 50, 47], joint audio-text foundation models [57, 13, 38], unsupervised speech recognition [4], discrete unit resynthesis [48, 19, 58], text-to-audio [32, 1, 17], and generative spoken language modeling [8, 26, 34]. Each of these methods operates on audio units exclusively greater than or equal to 25Hz, which has been a frequently cited area for future work to improve on [26]. Recent work [22] has also explored training speech encoder models with coarser units as targets.

**Extracting Semantic Units from Raw Data**    Also relevant to our work are several approaches, particularly in vision and audio, that generate emergent semantic clusterings from self-supervised transformer [53] models. In particular, the DINO approach in Caron et al. [10] observes object representations in attention maps through student-teacher distillation. Similar techniques have been also applied to audio to discover emergent syllable boundaries [12, 46]. These behaviors can vary heavily with small changes in pretraining strategy as explored in Darcet et al. [18]. Merging similar features has also been shown to produce significant vision model speedups such as in Bolya et al. [7]. Most similar to our work, Algayres et al. [2] extracted coarse continuous representations for GSLM, however these results trail behind NCLM-based approaches.

# 3    Learning Self-Supervised, Syllable-Like Representations from Raw Speech

In this section, we describe the bootstrapping process by which we extract low-bitrate speech units. We first describe LossPred, our algorithm to analyze outputs of self-supervised speech model loss functions to generate initial unit boundaries. Following this, we define SylBoost, an agglomeration procedure to iteratively refine these boundaries with student-teacher distillation. We also propose a new algorithm for the efficient extraction of boundaries from feature self-similarity matrices to fix the bottleneck slowing down VG-HuBERT and SD-HuBERT extraction.

## 3.1    LossPred: Extracting Syllable-like Segmentation from Relations in HuBERT's Loss

HuBERT has previously been shown to learn phone-like units with its K-means clusterings [28] which have formed the basis of subsequent works on GSLM and unsupervised ASR [4, 34, 26]. However, other work [42, 43] has shown that the representations learned by these models also correlate with higher level structure such as words, despite these structures not immediately appearing during clustering. Our goal in this section is to propose a method that can be applied to a pre-trained HuBERT model in order to automatically extract unit boundaries at the level of syllables or words, rather than phones. Although we apply our method to HuBERT, we expect that it could also be applied to other SSL speech models that utilize a similar loss function such as WavLM [11] or wav2vec2.0 [3]. The crucial commonality between these models is that they all utilize a masked language modeling (MLM) training objective, whereby input speech tokens are randomly masked and the model is trained to predict the masked inputs conditioned on the unmasked inputs.

We ground our intuition with the following thought experiment: If the input tokens corresponding to an entire word were replaced with mask tokens, we would expect the HuBERT model loss at these timesteps to be relatively high, as HuBERT would have to jointly predict word identity and the underlying acoustics to predict the missing span. On the other hand, if only the latter portion of a word were masked out, infilling this masked region given the word prefix may be easier by comparison. With this, if we iteratively shift a contiguous mask over a span of tokens and look at the loss, we would suspect to see a strong decrease in the loss throughout the timesteps corresponding to a masked semantic unit (word, syllable, or otherwise) as the beginning or end of the unit was partially revealed to the model. In our experiments, we find that semantic units extracted by this method tend to be syllable-like (both via inspection, and also confirmed experimentally in our segmentation and clustering experiments) and so we focus on these units for the rest of the paper.

We consider the setting of having a pretrained HuBERT teacher model and a HuBERT student model trained to predict the quantized contextualized representations generated by the teacher at layer $L$, as described in Hsu et al. [28]. Formally, given an input waveform $W$, we extract the teacher labels used to train the student by passing $W$ unmodified into the frozen HuBERT teacher and then quantizing the contextualized representations of layer $L$ with K-Means. We denote these teacher labels as $Y_{\{1...T\}}$, where $T$ is the number of tokens outputted by the CNN feature encoder stage of HuBERT. During pretraining, the student model is given a corrupted version of $W$ where tokens after CNN extraction at select times are replaced with a learned 'mask' embedding. We denote these tokens input to the

student as $X^M_{\{1...T\}}$ where $M = \{t_1, \ldots t_m\}$ is a contiguous span of masked timesteps. The student is then trained to predict these teacher labels at masked timesteps using a cross-entropy loss, which we denote as $E_t^{X^M}$ for the loss on $Y_t, t \in M$ given $X^M$:

$$E_t^{X^M} := -\log p(Y_t \mid X^M) \tag{1}$$

We look at the losses of the student model at the end of pretraining, and define the loss prediction matrix $C$ with mask span size parameter $s$ to capture the raw probabilities of the losses that would result from all possible temporal locations of the mask span $M$:

$$C_{r,c} \in \mathbb{R}_+^{T \times T} = \begin{cases} p(Y_t \mid X^M) \mid M = \{r+1, r+2, \ldots r+s\} & \text{if } r < c, |r-c| \leq \lfloor \frac{s}{2} \rfloor, \\ p(Y_t \mid X^M) \mid M = \{r-1, r-2, \ldots r-s\} & \text{if } r > c, |r-c| \leq \lfloor \frac{s}{2} \rfloor, \\ 0 & \text{otherwise.} \end{cases}$$

We separately calculate the upper and lower triangles of $C$, relating to the observed waveform being before the mask and after the mask respectively. In the upper triangle, each entry $C_{r,c}$ at row $r$ column $c$ is equal to $p(Y_t \mid X^M)$ given that the mask span in $X^M$ starts just after time $r$. Inversely, in the lower triangle, $C_{r,c}$ is equal to $p(Y_t \mid X^M)$ given that the mask span in ends just before time $r$. We use a span size $s = 50$ corresponding to 1 second as this duration is long enough to mask the majority of spoken words, and calculate the upper triangle based on the first 25 tokens of the mask span, and the lower triangle based on the last 25. We choose to use a span of tokens instead of masking all information after a timestep to prevent global information such as speaker information available to the model changing with respect to mask location. However, this limits us to only generating a diagonal span of probabilties as seen in 1. To extract $k$ regions with boundaries $B = \{b_1 = 1 < b_2 < \ldots < b_k = T+1\}$ from $C$, we adopt the min-cut algorithm discussed in Peng et al. [46], treating $C$ as the input feature-similarity matrix:

$$B := \underset{\{b_1=1<b_2\ldots<b_{k+1}=T+1\}}{\arg\min} \sum_{t=1}^{k} \frac{\sum\limits_{i=b_t}^{b_{t+1}-1} \sum\limits_{j=1}^{T} (C_{i,j} + C_{j,i}) - 2\sum\limits_{i,j=b_t}^{b_{t+1}-1} C_{i,j}}{\sum\limits_{i=b_t}^{b_{t+1}-1} \sum\limits_{j=1}^{T} (C_{i,j} + C_{j,i}) - \sum\limits_{i,j=b_t}^{b_{t+1}-1} C_{i,j}} \tag{2}$$

By choosing $k$ to be proportional to the length of the utterance, we can control the sample rate of our boundaries. We explore modifying this parameter in-depth throughout our experiments.

LossPred is expensive to run due to having repeat forward passes for sliding windows. To make this efficient, we extract multiple masked spans simultaneously with a gap between spans of three seconds. This results in roughly 200 forward passes of the student model to calculate $C$ on an arbitrarily-sized audio. We also preprocess the audio using a unsupervised voice activity dection model [51].

### 3.2 SylBoost: Bootstrapping Pesudo-Syllabic Units with Iterative Distillation

Given the initial boundaries predicted by LossPred, we follow the paradigm of noisy-student-teacher learning [55] to iterate and extract better representations. Our goal is to "sharpen" the syllabic organization in the feature space of an input student model that initially results from LossPred, as seen on the right of Figure 1. We choose a pretrained HuBERT [28] or Data2Vec2 [6] to initialize our student and teacher models, with the teacher model parameters held constant.

For a set of hypothesized speech segment boundaries $B = \{b_1 = 1 < b_2 < \ldots < b_{k+1} = T+1\}$, we group together all temporal tokens between two boundaries into disjoint groups $G_i = \{t \mid b_i \leq t < b_{i+1}\}$. For notation, we let $H_t$ map from $t$ to its corresponding group: $t \in G_{H_t}$. We apply our loss to the features at layer $L$, which we select based on syllabic correlation as explored in detail in Pasad et al. [43]. This results in student features $X^{(L)}_{\{1...T\}} \in \mathbb{R}^d$ and teacher features $Y^{(L)}_{\{1...T\}} \in \mathbb{R}^d$ where $d$ is the feature dimension.

Then the loss, which is applied to each token of the student model, is the mean squared error between the student features $X_t^{(L)}$ and the mean of the teacher features in the token's corresponding group:

$$Z := \frac{1}{T} \sum_{t=1}^{T} \left( X_t^{(L)} - \frac{1}{|G_{H_i}|} \sum_{s \in G_{H_i}} Y_s^{(L)} \right)^2 \tag{3}$$

This results in a model with a mean-squared-error feature similarity matrix as depicted in the right side of Figure 1. We then extract boundaries using a cut algorithm described later in Sec. 3.3, although the cut algorithm from Peng et al. [46] also works. With this, we can generate new pseudolabels and iterate the process again to extract better boundaries, which we perform twice.

### 3.3  Efficient Extraction of Unit Boundaries with SylBoost

To extract boundary indices from learned feature representations Peng et al. [46] proposed adapting the mincut approach in Malioutov and Barzilay [35]. However, for speech this approach is slow in practice and difficult to parallelize, bottlenecking our ability to extract boundaries in bulk across the large corpora necessary for downstream language modeling. Inspired by the SylBoost objective, we propose a more efficient approach for extraction: given $k + 1$ potential boundaries, we seek to choose groups that minimize the sum of the distances from each unit to the mean of its assigned group:

$$B := \underset{\{b_1=1<b_2...<b_{k+1}=T+1\}}{\arg\min} \sum_{i=1}^{k} \sum_{j=b_i}^{b_{i+1}-1} \left( X_j^{(L)} - \frac{1}{b_{i+1} - b_i} \sum_{l=b_i}^{b_{i+1}-1} X_l^{(L)} \right)^2 \qquad (4)$$

We further restrict the setting by choosing a maximum group length of $G$ tokens, where we choose $G = 50$ to correspond to one second of tokens, as syllables or words longer than this are fairly rare. With this, we can then split our algorithm into 1) calculating a distance array $D \in \mathbb{R}^{T \times G}$, where $D_{t,g}$ is the cost of the group of length $g$ ending at token $t$ and then 2) solving the minimal interval cover from this distance array with dynamic programming. An efficient implementation using PyTorch [44] on CUDA [40] runs in $O(k)$ data-aware sequential steps.

## 4  Syllable-LM: Speech Unit Language Modeling Over Syllable-Like Units

### 4.1  Language Model

GSLM [34] defines a pipeline for modeling raw audio as three stages: 1) Audio-to-unit Tokenization, 2) Running a decoder transformer model on these units, and 3) Decoding the tokens back into a waveform. Like AudioLM and TWIST, we use an autoregressive transformer decoder language model to approximate $p(x_t \mid x_{t-1}, \ldots, x_1)$ given an input token sequence $x_1, \ldots, x_T$. We refer to this model as SpeechLM. We train it on clusters which we extract by mean pooling features at layer $L$, chosen as before, over their boundary groups, followed by K-Means and Agglomerative Clustering to a desired number of discrete units. Like TWIST, we prepend a <BOS> token and make no other special changes. Due to current prevalence of this architecture, we refer to [26] for additional details.

### 4.2  Resynthesis and the Vocoder

For resynthesis, we adopt the interleaved decoding strategy from Song et al. [50] to output the mHuBERT units from TWIST [26], obtaining a waveform by cascading this output into their provided mHuBERT-to-speech vocoder. This interleaving strategy demonstrates superior performance in high-difficulty settings compared to other Neural Codec Lanauage Models like VALL-E [54], and so we use it for all resynthesis experiments. Although the cascading procedure may produce additional errors, we choose this approach for the following reasons:

1. Text-to-speech systems like VALL-E traditionally start by converting text units into phones using rule-based strategies to improve quality. This indicates that traditional unit-to-speech resynthesis methods might be challenging for our low-bitrate units.
2. This pipeline allows for fast experimentation as we can precompute the mHuBERT 25hz units once for all training runs.
3. Using the same Vocoder allows for fairer comparisons against TWIST.

To interleave our units, we sort on the start-timestep of every pseudo-syllable unit and mHuBERT-unit in ascending order. To decrease the odds of mHuBERT units appearing before the pseudo-syllable unit corresponding to the same ground truth syllable due to errant SylBoost boundaries, we subtract 0.08s (the length of two mHuBERT frames) from each pseudo-syllable start time before sorting. For the rest of the pipeline, we follow [50] with our syllables as a drop-in replacement for phones. We note

Table 1: Unsupervised Syllable Boundary Detection and Clustering Accuracy on LibriSpeech [41] Test. For F1 scores, the superscript is tolerance threshold in ms. All other metrics use 50ms. Higher is better.

| Approach | Backbone | Training | F1$^{50}$ | F1$^{20}$ | Pr. | Re. | R | CPur | SPur |
|---|---|---|---|---|---|---|---|---|---|
| Feat-Sim[43] | HuBERT | no | 47.3 | 24.7 | 46.6 | 48.0 | 54.5 | 28.0 | 30.0 |
| LossPred (Ours) | HuBERT | no | 59.6 | 31.4 | 54.9 | 66.7 | 56.3 | - | - |
| SD-HuBERT[12] | HuBERT | yes | 66.1 | 32.2 | 64.9 | 67.4 | 70.7 | **43.2** | 45.0 |
| SylBoost (Ours) | HuBERT | yes | 70.9 | 40.1 | 70.8 | 71.4 | 75.1 | 28.9 | 47.8 |
| SylBoost (Ours) | Data2Vec2 | yes | **73.2** | **44.6** | **72.1** | **74.4** | **76.9** | 33.6 | **54.9** |

that although our interleaved resynthesis model slows down generation compared to TWIST, most model parameter scaling happens in the SpeechLM. For example, the TWIST paper still observes scaling improvements at 13B parameters while current SOTA TTS models such as [29] operate well with fewer than 1B parameters.

We then generate continuations for a sample by 1) Extracting syllable-unit and mHuBERT units from the sample, 2) Sampling syllable-unit continuations from the SpeechLM, 3) Continuing mHuBERT units with our interleaved model conditioned on sample mHuBERT units, sample syllable-units, and continued syllable-units, and 4) Resynthesizing these into speech using the vocoder.

## 5 Experiments

### 5.1 Training Datasets

We train our tokenizer using LibriSpeech [41], which contains 960 hours of audio books. We noticed that the agglomeration procedure described in 3.2 converges before all data is used, and so we randomly subsample LibriSpeech to a 100 hour train set and train for five epochs and two iterations for all experiments. We train our SpeechLMs using all of LibriLight [30], which provides roughly 55k hours of speech. As a note on fair comparison, although AudioLM uses exactly this split of LibriLight, TWIST collects an additional 100k hours of data, totaling to 155k hours.

### 5.2 Model Details

We implement using the OPT [59] flavor of models and default to using 12 layers, an embedding dimension of 768, and learned positional embeddings for both our SpeechLM and our Interleaved-Vocoder-LM. This totals to 90M non-embedding parameters, the same as TWIST-125M. We also experiment with a larger 24 layer 1024 dimension model totaling to 300M non-embedding parameters, the same as AudioLM and TWIST-350M. For all pretraining experiments we randomly crop files to 25 seconds, use a batch size of 80000 tokens, and train for 200k steps, which amounts to the same compute as in TWIST. To make our approach entirely textless, we do not use TWIST initialization. Additional hyperparameters and hardware details are in Appendix A.2.

### 5.3 Tokenizer Experiments

By varying the number of boundaries input to our cut algorithm at each stage in the agglomeration pipeline, we can arbitrarily control our rate of temporal tokenization. We evaluate three main unit-rates at 8.33Hz, 6.25Hz, and 5.00Hz, the latter which matches the empirical rate of SD-HuBERT units on LibriSpeech dev-clean. Combining unit-rates with changing the number of clusters generated by K-Means and Agglomeration gives us fine-grained control of the model bitrate. We note that although SD-HuBERT applies a cut algorithm, this is done after thresholding low-magnitude features that emerge from pretraining. As a result, we find that we cannot control the frequency of SD-HuBERT units by changing parameters of its mincut algorithm becuase additional cuts result in close-to-identical representations that map to the same quantized clusters.

From prior work, we compare against the AudioLM tokenizer w2v-BERT [14], and the tokenizer from TWIST which is an open-source HuBERT model pretrained for an additional iteration on a large and diverse set of multilingual data, henceforth mHuBERT. Both of these tokenizers operate at 25Hz

Table 2: Unit Resynthesis. WER/CER results on 4-10 second examples on LibriSpeech [41] test-clean. Hz and Bitrate are measured post Run-Length-Encoding (RLE) on LibriSpeech dev-clean.

| Model | Changes | Hz | #Units | BPS | WER↓ | CER↓ |
|---|---|---|---|---|---|---|
| SD-HuBERT [12] | | 5.0 | 4096 | 60 | 37.3 | 22.7 |
| SylBoost (HuBERT) | +Our Clustering | 5.0 | 4096 | 60 | 18.5 | 10.2 |
| SylBoost (D2V2) | +Use Data2Vec2 | 5.0 | 4096 | 60 | 12.8 | 6.4 |
| SylBoost (D2V2) | +Increase #Units | 5.0 | 16384 | 70 | 9.1 | 4.3 |
| SylBoost (D2V2) | +Tune unit-rate, #Units | 8.33 | **2048** | 91 | 8.0 | 3.7 |
| SylBoost (D2V2) | +Tune unit-rate, #Units | 6.25 | 8192 | 81 | **7.0** | **3.2** |
| mHuBERT (upper bound) [26] | | 19.5 | 500 | 175 | 6.3 | 2.5 |

Table 3: Main SyllableLM results. We evaluate on sWUGGY (In-Vocab, All, Out-of-Vocab), sBLIMP from ZeroSpeech [39], and tStoryCloze from Hassid et al. [26]. Higher is better. *Estimated.

| Model | Params | #Units | Hz | BPS | #Data Toks | GPU-Hours | sWUGGY All | IV | OOV | Semantics sBL. | tSC |
|---|---|---|---|---|---|---|---|---|---|---|---|
| Phone Topline | 90M | 70 | 12.5 | 76 | 2.5B | 70 | 81.4 | 95.2 | 67.7 | 68.8 | 80.6 |
| Syllable Topline | 90M | 28k | 5.0 | 74 | 1B | 70 | 79.5 | 93.1 | 65.9 | 69.3 | 76.6 |
| AudioLM [8] | 300M | 1k | 25 | 250 | 5B | 2.9k* | 71.5 | **83.7** | 59.3 | **64.7** | - |
| TWIST [26] | 300M | 500 | 19.5 | 175 | 9B | 295 | 70.6 | 80.3 | 61.0 | 56.2 | 69.9 |
| TWIST | 1.3B | 500 | 19.5 | 175 | 9B | 1.1k* | 71.8 | 81.1 | 62.3 | 57.0 | 70.6 |
| TWIST | 7B | 500 | 19.5 | 175 | 9B | 5.9k* | 72.7 | 83.6 | 61.8 | 59.0 | 74.1 |
| TWIST | 13B | 500 | 19.5 | 175 | 9B | 10k* | 73.9 | 84.1 | 63.7 | 59.2 | 76.4 |
| TWIST-CI | 90M | 500 | 19.5 | 175 | 3.9B | 84 | 69.7 | 79.8 | 59.7 | 55.5 | 69.0 |
| BPE [49] | 90M | 4k | 9.8 | 118 | 2B | 84 | 61.8 | 66.7 | 56.8 | 54.5 | 56.2 |
| SyllableLM | 90M | 2k | 8.3 | 91 | 1.6B | **70** | **72.2** | 81.7 | **62.6** | 62.4 | 71.4 |
| SyllableLM | 90M | 8k | 6.25 | 81 | 1.2B | 75 | 72.1 | 82.2 | 61.9 | 62.9 | 70.2 |
| SyllableLM | 90M | 16k | **5.0** | **70** | **1B** | 82 | 67.6 | 76.9 | 58.3 | 63.2 | 69.0 |
| SyllableLM | 300M | 8k | 6.25 | 81 | 1.2B | 290 | **72.2** | 82.2 | 62.0 | 63.7 | **75.4** |

followed by Run Length Encoding, which deduplicates repeated units. We additionally reimplement Byte Pair Encoding as done in Shen et al. [49] on the deduplicated mHuBERT units, resulting in the lowest bitrate encoding of speech outside of our model. We grid search and find that the minimum bitrate from BPE is obtained from 4k-16k units and choose 4k units for all experiments (Shen et al. [49] originally operated on 50Hz units, meaning that the 117bps rate obtained here is also new).

Because we want to use a 50Hz base encoder to match SD-HuBERT and have fine-grained boundary control during syllable segmentation, we cannot use the 25hz mHuBERT encoder from TWIST. Unfortunately, this means that the quality of the base encoder may be a confounding factor in our SpeechLM evaluation. We choose Data2Vec2-base [6] as a middleground for training SpeechLMs on syllable-like units because we find its quality enables lower bitrates than HuBERT, but it is older and trains on less-data than mHuBERT from TWIST, and it has 6x fewer parameters than w2v-BERT, used by AudioLM. We suspect that applying newer encoders like w2v-BERT 2 from Communication et al. [16] could enable even better performance, which we leave to future work. We initialize Data2Vec2 SylBoost from the same HuBERT loss boundaries as discussed in 3.1.

## 5.4 Results: Evaluating Unit Quality

We evaluate the quality of our semantic units with two approaches 1) measuring correspondence with syllables and 2) running speech resynthesis followed by ASR. To measure correspondence with syllables, we use the development and test sets of LibriSpeech [41] and follow the approach from Peng et al. [46], extracting timesteps for phones on using the Montreal Forced Aligner [36] and then converting these phones into syllables with a rule-based method [25]. We evaluate the quality of syllable boundary detection with a ground truth boundary marked as hit if a proposed boundary is present within a tolerance threshold. We report F1, Precision, Recall, and R score. We ablate F1 scores with tolerance windows of 20ms and 50ms. Given boundaries, we also evaluate the purity of

Table 4: Boundary detection with different initialization using HuBERT on LS dev-clean

| Model | F1 | Pr. | Re. |
|---|---|---|---|
| Similarity | 46.7 | 48 | 45 |
| -Iter 1 | 51.1 | 50 | 52 |
| -Iter 2 | 50.4 | 51 | 50 |
| Loss-Corr | 60.1 | 53 | 68 |
| -Iter 1 | 67.1 | 67 | 68 |
| -Iter 2 | **70.2** | **70** | **70** |

Table 5: Controllability of unit rate measured on LibriSpeech dev-clean boundary detection. D2V2, 50ms threshold. P:Phone, S:Syllable, W:word

| Hz | F1-P | F1-S | F1-W |
|---|---|---|---|
| 8.33 | **72.0** | 63.5 | 56.8 |
| 6.25 | 65.2 | 71.8 | 66.0 |
| 5.0 | 58.7 | 73.0 | 71.8 |
| 4.3 | 54.3 | **73.2** | **74.0** |

Table 6: Holding number of units and unit rate constant. ZeroSpeech development set.

| Hz | #T | sWU. | sBL. |
|---|---|---|---|
| 8.33 | 4k | **72.9** | 61.8 |
| 6.25 | 4k | 69.3 | **63.3** |
| 5.00 | 4k | 65.7 | 62.8 |
| 8.33 | 2k | 72.1 | **62.0** |
| 8.33 | 4k | **72.9** | 61.8 |
| 8.33 | 8k | **72.9** | 61.2 |

our clusters with 4096 units, with Syllable Purity measuring the probability that a syllable is mapped to its most corresponding cluster unit, and Cluster Purity measuring the probability that a cluster is mapped to its most corresponding syllable unit.

Even if units do not correspond with syllables, they can still be of great use to SpeechLMs if they can resynthesize back into speech that matches the original text. Additionally, training a resynthesis model provides a stronger description of the semantic information contained in units than purity metrics, which are especially problematic because SD-HuBERT does not provide a unit at every timestep while our methods do, possibly making cluster and syllable purity evaluation unreliable. To evaluate resynthesized speech, we follow AudioLM and measure Word Error Rate (WER) and Character Error Rate (CER) on the set of 4-10 second segments from LibriSpeech test-clean. For ASR, we follow VALL-E [54] and use the public HuBERT-base CTC ASR model provided by [28].

Table 1 shows our syllabic correspondence results against the prior-state-of-the-art SD-HuBERT [12] and the HuBERT-based feature similarity strategy from [46]. Applying our LossPred followed by agglomeration strategy on either HuBERT or Data2Vec2 improves performance across-the-board except for in cluster purity. Although it LossPred SD-HuBERT in performance, it pushes the boundary for syllable recognition using HuBERT without additional training. We justify using LossPred as a bootstrapping source instead of a HuBERT similarity metric [46, 43] in Table 4, which we discuss more in Appendix A.4. Improvement across iterations and with different loss initialization can be found in Table 4. We explore the effects of changing the unit rate on boundary predictions in Table 5.

We compare against prior SpeechLMs and demonstrate the step-by-step changes used to improve unit cluster re-synthesis quality as compared to SD-HuBERT in table 2. We observe over a 50% decrease in WER and CER by applying our method using the SD-HuBERT base parameters. We further decrease WER by a third by using Data2Vec2, and from there by modifying the unit sample rate and number of clusters can reach as low as 2048 clusters and a WER of 7%. These results demonstrate by far the lowest bitrate we are aware of for 'reasonable-quality' self-supervised-unit resynthesis. Resynthesis for all models we train is done back into mHuBERT-25Hz units, bounding potential quality at a WER of 6.3%.

## 5.5 Results: Generative Spoken Lanauage Modeling

The end-to-end GSLM pipeline is deep, and so it is essential to have metrics to independently evaluate different stages. To evaluate our SpeechLM stage, we follow Lakhotia et al. [34] and use the ZeroSpeech [39] sWUGGY and sBLIMP evaluation. The sWUGGY dataset tasks the model with outputting a higher perplexity on similar but fake spoken words (e.g. brick vs blick). Similarly, the sBLIMP dataset checks syntactic correctness (e.g. the dog sleeps vs the dogs sleeps). We also evaluate the SpeechLM on the tSC set from Hassid et al. [26], which operates like the ZeroSpeech metrics on a spoken version of the StoryCloze dataset [37] with the last sentence in negative samples randomly chosen. For all metrics we follow prior work and output the mean perplexity per token.

The results for SpeechLM metrics are in 3. We reimplement a 90M parameter model using the TWIST mHuBERT units without textually-pretrained initialization (Cold-Init in the TWIST paper) on our data split for an all-else-held equal comparison on unit type. We also train on BPE units as described in 5.3, the next-lowest bitrate units outside of our model. For textual toplines, we train on corresponding LibriLight text transcripts from Kang et al. [31] and convert text to phones and

syllables using the same methods as in Section 5.4. We find that training with each of our syllable units improves perfromance across-the-board on sBLIMP and tSC versus comparably-sized models and is competitive against larger models. In fact, with under 90 hours of training, SyllableLM outperforms even the 13B parameter TWIST on sBLIMP. We also beat AudioLM on the full split of sWUGGY with 30x less GPU compute and TWIST model sizes up to 1.3B parameters. On tSC, we observe that SyllableLM large approaches performance of the textual topline, outperforming all models except for TWIST 13B. Due to compute requirements, we are unable to scale further.

We notice a decrease in sWUGGY quality with our 5.0Hz units, which we suspect is in part caused by the short length of the dataset audios making input tokenization excessively short. We further ablate these differences in table 6. We also find that BPE, despite having the lowest bitrate outside of our approach, does not approach the quality gains created by our syllable-like units.

Table 7: Continuation Metrics. We measure PPX@Oracle-VERT and VERT@Oracle-PPX as implemented in Lakhotia et al. [34]

| Model | PPX@O-V | VERT@O-P |
|---|---|---|
| TWIST 300M | 205 ±24 | 24.0 ±1.0 |
| TWIST 1.3B | 175 ±14 | 22.6 ±1.2 |
| 8.33Hz 2k 90M | 159 ± 8 | **15.1** ±0.9 |
| 6.25Hz 8k 90M | 139 ±12 | 20.1 ±0.7 |
| 5.00Hz 16k 90M | 131 ±11 | 15.2 ±1.0 |
| 6.25Hz 8k 300M | **116** ± 7 | 15.8 ±0.9 |

To measure the quality of end-to-end continuations, we use the VERT@O-PPX and PPX@O-VERT metrics proposed in Lakhotia et al. [34], which are shown to be the automatic metrics correlating best with human meaningfulness judgements. VERT@O-PPX measures the diversity of output at the sampling temperature where the perplexity of generated audio transcriptions matches that of the ground truth text, and PPX@O-VERT performs the inverse. Like Lakhotia et al. [34], we generate 10-second continuations from 1000 randomly sampled 3-second crops from LibriSpeech test-clean, and measure results using their provided environment and parameters. We report these in Table 7 with two sigma error bars, outperforming TWIST 300M and 1.3B.

## 6   Limitations

Though speech is a very general medium, there are a number of challenges in adapting our methods to generate low-bitrate units angled towards other audio tasks or other domains such as vision. Our LossPred technique assumes that the semantic units to learn are separable across time, one-dimensional, and contiguous. In audio tasks or settings with multiple speakers, sounds or words can occur simultaneously and can't be separated across the time dimension. Images and video are multi-dimensional, not allowing a trivial sliding window approach. Images and video can also have partially occluded or overlapping objects, violating continuity. Furthermore, it is still unclear whether our longer units will be better at scaling to larger datasets, such as the 4.5M hours used by Communication et al. [16]. For example, our semantic units may be losing out on useful paralinguistic features like tone whose impact is only salient on non-audiobooks or at scale. It is also important to note that large textual language models can have harmful effects, such as enabling the generation of misinformation in mass. Although generative spoken language models have not yet caught up to their textual counterparts, it is still necessary to be aware of potential misuses that could arise in the future.

## 7   Conclusion

We introduce a new method to tokenize speech for use in GSLMs. We do this by proposing a method to elicit syllabic organization in pretrained speech encoder models, bootstrapping a feature-space agglomeration algorithm from a static analysis of correlations in off-the-shelf teacher and student model losses across time. We demonstrate the success of our technique both in having strong associations with syllables and as an extremlely low-bitrate codec for speech resynthesis. Using this tokenization strategy, we successfully train SyllableLM, a SpeechLM that out-performs comparable state-of-the-art approaches across a diverse range of metrics with a significant inference speedup. We further ablate several design decisions such as quantitization strategy, loss initialization, and the effects of controllability for downstream usecases. Compression is a crucial aspect of learning, and we hope that these significant improvements in the unsupervised learning of low-bitrate speech units can serve as a foundation for approaches towards understanding spoken language and general representation learning.

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

# A    Appendix / supplemental material

## A.1    Randomly Sampled Example Segmentations

We provide randomly sampled example segmentations from the LibriSpeech [41] dev-clean set.
All models are the second iteration of Data2Vec2, which we use for our SyllableLM experiments
in Section 5.5. Top: Feature Self-Similarity matrix, darker green is closer. Segmented cuts span
vertically in blue from the top, ground truth boundaries span vertically in red at the bottom. Bottom:
time-aligned Mel-Spectrogram. We call attention to the interesting behavior of global correspondences
appearing when words or syllables are repeated. Best viewed zoomed in.

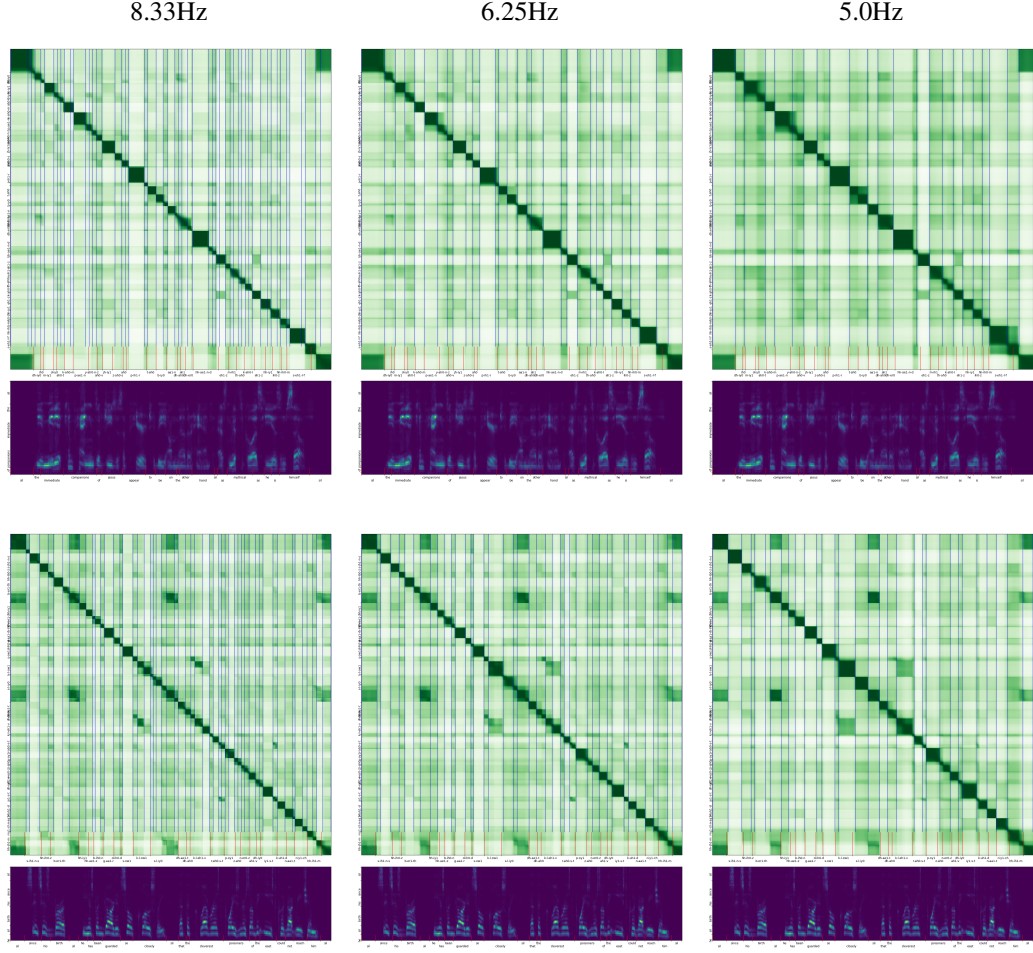

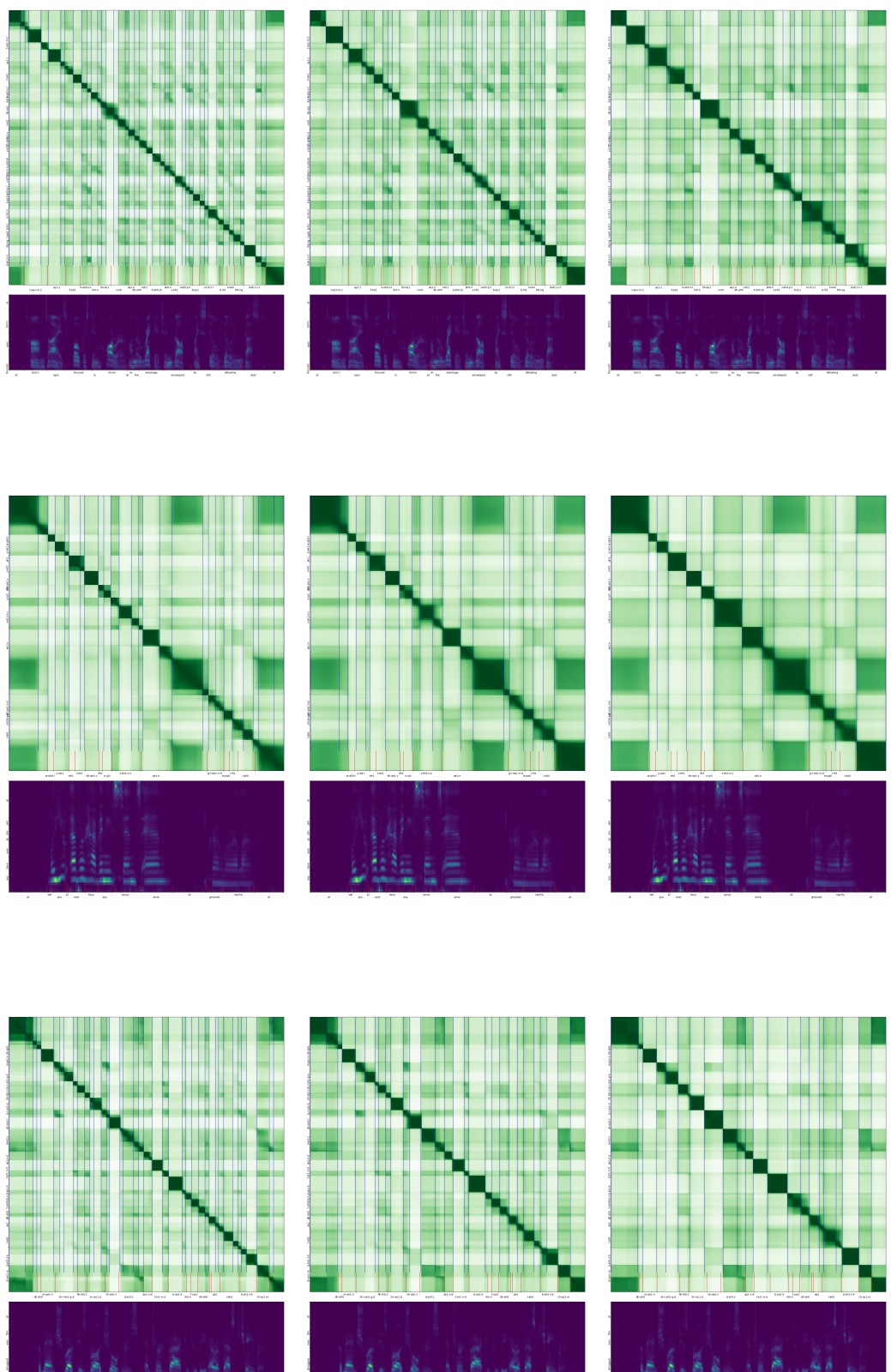

## A.2 Hardware And Hyperparameters

We implement all experiments using NVIDIA A40 46GB GPUS with a Intel Xeon Gold 6226R CPU @ 2.90GHz. Estimated speeds are made using these results as well as scaling from Zhang et al. [59].

Hyperparameters for pretraining our models are below. We note that the Batch Size is in terms of tokens, which means that higher unit rates will have fewer seconds of raw audio per batch to keep GPU compute roughly equal per model.

Table 8: Speech pre-training hyper-parameters.

|  | SyllableLM Base | SyllableLM Large |
|---|---|---|
| Layers | 12 | 24 |
| Embed Dim | 768 | 1024 |
| MLP Dim | 3072 | 4096 |
| GPUs | 2 | 4 |
| Learning rate | $2 \times 10^{-4}$ | $2 \times 10^{-4}$ |
| Adam $\beta_1$ / $\beta_2$ | 0.9 / 0.98 | 0.9 / 0.98 |
| Weight decay | 0.01 | 0.01 |
| Learning rate schedule | Linear Decay | Linear Decay |
| Dropout | 0.1 | 0.1 |
| LayerDrop | 0.0 | 0.0 |
| Warmup updates | 8,000 | 16,000 |
| Batch size (tokens) | 80,000 | 80,000 |
| Updates | 200,000 | 200,000 |
| Position Embeddings | Learned | Learned |

## A.3 Speedup

Table 9: Inference speed results, measured in Real-Time-Factor, the processed seconds per second. We use 32 Batches with 25 seconds of audio each, which matches the length of our training data. 1 GPU, 16 Cores. Standard error less than 1 sec/sec

| Encoder | Real-Time-Factor ↑ |
|---|---|
| SD-HuBERT [12] | 368 |
| HuBERT+MinCut [46] | 88 |
| HuBERT+MinSum 3.3 | **488** |
| SpeechLM, 100M, Cached Units | |
| TWIST | 7.8k |
| Ours 6.25Hz 8k | **34.7k** |

## A.4 Discussion: Other Bootstrapping Strategies

Of course, there already exist several strategies for unsupervised syllable and word segmentation such as Fuchs and Hoshen [23] and Pasad et al. [43] that could be used to bootstrap our first pseudolabels. We find however in our experiments that these approaches, which are calculated using the similarity or dissimilarity of HuBERT embeddings across time, converge to a lower quality in bootstrapping than our proposed method. We suspect that this may be caused by the fact that although the representations of these models correlate with boundaries, there is no modeling in the pretraining loss pushing the representations to linearly separate across semantic differences. Meanwhile, the loss is forced to change across semantic boundaries due to the difficulty of language modeling, albeit noisily.

## A.5 Sample Continuations

Below are sample continuations generated with temperature sampling parameter chosen to best match Oracle VERT diversity scores. We provide continuations of roughly 3 seconds of audio, sampled randomly from LibriSpeech test-clean. This text is given as output by our HuBERT ASR Model from [28], with transcription errors present and with no additional modifications. The source text is bolded, and sometimes cuts off mid-word, which can behave differently per sample based on unit rate and quantization artifacts.

| Model | Continuation |
|---|---|
| TWIST 300M | **HE FILLED FOR AND FOUND THE WIZARD'S BLACK CLOTH THE SQUIRE WAS** QUITE DEVOTED TO CARRYING IT IN THE PARKS I COULD TELL YOU THAT HE RATHER LOVED ALL THE WIZARDS IN THE MATTER OF HONEST FIGUROUS TEMPERED MEN THEN IMMEDIATELY AND BECAUSE HE DID |
| TWIST 1.3B | **HE FILLET FOR AND FOUND THE WIZARD'S BLACK CLOTH THE SQUIRE WAS** QUICK IN MOVEMENT A WHEEL OF HIS JOURNEY MADE HIM THE IMPRESSION OF HARE'S BLOODSHED THERE WAS NO HELP FOR IT HE SURMISED AND HE DID SO |
| SyllableLM 2k 8.33Hz 90M | **HE FILLED FOR AND FOUND THE WIZARD'S BLACK CLOTH THE SQUIRE WAS** GLAD THAT HIS LAD HAD NOT RETURNED TO KING LENO FOR IT IS PROBABLE THAT HE WOULD HAVE BECOME SO UNWELL THAT HE COULD NOT HAVE SPOKEN |
| SyllableLM 8k 6.25Hz 300M | **HE FILT FOR AND FOUND THE WIZARD'S BLACK CLOTH THE SQUIRE WAS** HALF WAY DOWN THE PATH WITH THE WIZARD BUT HE STOPPED SHORT BEFORE THE GATE CLUTCHED HIS FISTS SEIZED HIS GOWN AND |

| Model | Continuation |
|---|---|
| TWIST 300M | **WHEN THEY WERE OUTSIDE UM SIMPLY LATCHED THE DOOR AND STARTED** UP THE STAIRS AND EVEN WENT INTO THE LITTLE HOUSE AROSE IN THE SETTING SUN SET FORTH THE PLACES WHERE PEOPLE LIVED FROM MON HONEYSUCKLE HANNEY |
| TWIST 1.3B | **WHEN THEY WERE OUTSIDE UM SIMPLY LATCHED THE DOOR AND STARTED** UP THE TURNPIKE OL HAT DON KILL ME THE SLING DE IN YORN ME WEAVING OUT CHARLEYS SENSE EXAMINED WHAT HE MADE EXAMINES |
| SyllableLM 2k 8.33Hz 90M | **WHEN THEY WERE OUTSIDE UM SIMPLY LATCHED THE DOOR AND STARTED** WALKING IN THEY WERE TOO OLD TO CARE MUCH ABOUT GOING HOME THEIR RELATIVES LEFT |
| SyllableLM 8k 6.25Hz 300M | **WHEN THEY WERE OUTSIDE UM SIMPLY LATCHED THE DOOR AND STARTED** SLOWLY DOWN THE CORRIDOR AND MISSUS BAKER WALKED BESIDE THEODORA THEY WERE NEAR THE OUTER DOOR WHERE |

| Model | Continuation |
| --- | --- |
| TWIST 300M | **DO BE OR NOT TO BE THAT IS THE QUESTION WHETHER TIS NO**BODY SIBL LINE IN OTHER SHIRTS OR CHOCOLATE NOS MICOTTON BUTTER WHAT WE WERE DO WE SEE THESE HITS WE'VE GOT THE GHOST HERE THEY'RE LOOKING |
| TWIST 1.3B | **DO BE OR NOT TO BE THAT IS THE QUESTION WHETHER TIS NO** GOOD EITHER THAN TO GO THROUGH THE JUDGMENT OF GAUL AND YOUR DOCTRINES THE LORD YOUR GOD AND YOUR GOSPEL IN RESPECT OF THE POWER OF THIS |
| SyllableLM 2k 8.33Hz 90M | **DO BE OR NOT TO BE THAT IS THE QUESTION WHETHER TIS NO** OTHER THAN ESO'S OWN DESTINY YOU SEE IT IS A LA MISTER PRIOR THAT THIS IS THE CASE |
| SyllableLM 8k 6.25Hz 300M | **DO BE OR NOT TO BE THAT IS THE QUESTION WHETHER TIS NO** REGRET OR NO PLEASURE THAT MAY BE RUSHED INTO ACTION AT ONCE WITH THE GREATEST EAGERNESS OF IMPULSE AND ELASTICITY OF HEART |

| Model | Continuation |
| --- | --- |
| TWIST 300M | **HE IS CALLED AS YOU KNOW THE APOSTLE OF THE INDI**AN KING WHO IS SO GLORIOUS AND ACTING WHY IS THE OTHER PRINCE NOT BELIEVED BY HIM IN EVERY FAITH THAT IS FREE WILL EXCEPT WHEN HE |
| TWIST 1.3B | **HE IS CALLED AS YOU KNOW THE APOSTLE OF THE INDI**ES SAW WHAT HAD PASSED THROUGH HIM LATER IN ANOTHER BOOK AMONG THOSE WHO HAD ENGRAVED IT THIS VOLUME MISTER PICKWICK THOUGHT IT RIGHT NOT TO INSULT YOU |
| SyllableLM 2k 8.33Hz 90M | **HE IS CALLED AS YOU KNOW THE APOSTLE OF THE INVID**ISIBLE BEFORE THEY RECEIVED THE GRACE OF GODAD CHRIST THEN HAD IN THE FAITH OF HIS SON |
| SyllableLM 8k 6.25Hz 300M | **HE IS CALLED AS YOU KNOW THE APOSTLE OF THE INDI**ES HE IS THE FORERUNNER OF TEACHING AND FAR BEYOND IT HE IS THE EXACT SCIENTIST WHO MEASURES THE MOVE |