# OpenReview forum: "SyllableLM: Learning Coarse Semantic Units for Speech Language Models"
_NeurIPS.cc/2024/Conference — Submitted to NeurIPS 2024_

### Official Review · Reviewer_kvPb · 2024-07-12

**Soundness:** 3
**Presentation:** 1
**Contribution:** 2
**Rating:** 3
**Confidence:** 4

**Summary:**

This paper proposes a two-stage speech language model with semantic tokens and acoustic tokens similar to AudioLM ([Borsos et al., 2022]).
-   The semantic tokens come from a speech tokenizer that can group a variable number of frames into a single token. To train such a speech tokenizer,
    1.  This paper first takes inspirations from syllable-like structures uncovered from HuBERT, and produces an initial segmentation (Section 3.1).
    1.  An iterative process is then applied to improve the segmentation (Section 3.2).
    1.  Finally the tokens are obtained by clustering of the mean-pooled frame features (Section 4.1).
-    The acoustic tokens are identical to the HuBERT-based tokens in ([Hassid et al., 2023]), referred to as "mHuBERT" in this paper.

Experiments with the proposed model demonstrate the following when compared to previous work,
-   Better unsupervised syllable segmentation
-   Lower speech reconstruction WER
-   Better or competitive accuracy in speech language modelling tasks (sWUGGY, sBLIMP, tStoryCloze) with lower compute
-   Better speech continuation quality

[Borsos et al., 2022]: https://arxiv.org/pdf/2209.03143 "AudioLM: a Language Modeling Approach to Audio Generation"
[Hassid et al., 2023]: https://proceedings.neurips.cc/paper_files/paper/2023/file/c859b99b5d717c9035e79d43dfd69435-Paper-Conference.pdf "Textually Pretrained Speech Language Models"

**Strengths:**

-   Originality: This paper proposes an original method for producing syllable-like segmentation of speech in an unsupervised manner.
    -   It uses conditional probabilities from a masked language model instead of feature similarity ([Peng et al., 2023]) to detect initial syllable boundaries.
    -   The use of an iterative process to further improve the segmentation quality is also original.
-   Quality: This paper is well-motivated. The experiment design is sound. Ablation studies included in the experiments provide valuable insight to various modelling choices.
-   Clarity: The experiment results are reported in an easy-to-interpret manner.
-   Significance: The proposed model is an competitive speech language model with a lower inference computational cost.

[Peng et al., 2023]: https://arxiv.org/pdf/2305.11435 "Syllable Discovery and Cross-Lingual Generalization in a Visually Grounded, Self-Supervised Speech Model"

**Weaknesses:**

I think the method and results in this paper would make a good paper for NeurIPS, however I cannot make a recommendation for acceptance because this paper needs substantial revision to improve its readability. A non-exhaustive list of issues making the paper hard to follow includes the following,
-   References to items not yet introduced
    -   Lines 153-154, the phrase "our loss" make it sound like a referrence to the masked language model loss discussed in the previous sub-section, whereas in fact it is referring to Equation (3), a yet-to-be-introduced loss for SylBoost.
    -   Lines 159-162 give a very vague description of the "similarity matrix" and the "cut algorithm" which can only be known if the reader has already seen the subsequent Section 3.3.
    -   Starting at line 188, Section 4.2 makes repeated references to "mHuBERT". "mHuBERT" appears to be name given to the acoustic tokens in ([Hassid et al., 2023]) by this paper (line 241). ([Hassid et al., 2023]) itself does not use this name, so an ordinary reader would not be able to tell what an "mHuBERT" model is when they work through Section 4.2.
-   Confusing terminology
    -   "pretraining": This paper makes a liberal use of the term "pretraining" to the point it's very difficult to tell which is the model being "pretrained". For example,
        -   Line 113 mentions a "pretrained HuBERT teacher model", then line 119 says "during pretraining, the **student** model ...". The teacher and the student are presumably not trained at the same time, yet the use of "pretraining" in this context make it appear that the contrary is happening.
        -   Line 225 says "for all pretraining experiments". A reader will have to look really closely to see this means "training of the speech LM", not "pretraining HuBERT, etc".
    -   "Agglomeration" vs "SylBoost": This paper appears to use these two terms interchangeably. Agglomerative clustering is apparently also used  (line 183). This makes it difficult for the reader to tell when "agglomeration" is mentioned, whether the authors intend to refer to SylBoost or just the clustering.
-   Confusing equation
    -   The unnumbered equation between line 126 and line 127 defines the similarity matrix from MLM probabilities. It makes reference to
$Y_t$ without specifying which $t \in M$ is used to define $C_{r,c}$. As a result, after having read the paper 6 times over, I still do not know how to compute $C_{r,c}$.
-   Writing style
    -   Overall the writing style of this paper is very wordy, inconcise and disorganized. Often the same message can get through with far shorter sentences. Most of the paragraphs read like a dump of the stream of consciousness of the author instead of a technical document intended for actual readers. For example,
        -   Lines 102-112 would be a lot easier to understand with formal notations and a concrete example.
        -   Lines 127-131 appear to be a mere repetition of the equation above, without any new information.
        -   Lines 242-255 contain a large amount of disorganized modelling details.

[Hassid et al., 2023]: https://proceedings.neurips.cc/paper_files/paper/2023/file/c859b99b5d717c9035e79d43dfd69435-Paper-Conference.pdf "Textually Pretrained Speech Language Models"

**Questions:**

-   Is LossPred really necessary?
    -   While Table 4 shows that LossPred is a better initialization strategy for SylBoost, would a cheaper initialization strategy (like feature similarity or even random) produce equally good segmentation given more iterations?
    -   Suppose the answer to the previous question is yes, would it make sense to run SylBoost but using the final activation of HuBERT instead of an intermediate layer (line 154)? The final activation may be better semantic features.

**Limitations:**

The authors adequately addressed the limitations.

---

> ### Author Rebuttal · Authors · 2024-08-05
>
> We would like to start by thanking kvPb for their clear and significant commitment to understanding our paper in detail. In our rebuttal below, we try to demonstrate that the presentation clarity errors pointed out only need minor changes to be fixed.
>
> **Regarding "Confusing equation":**
> > * The unnumbered equation between line 126 and line 127 defines the similarity matrix from MLM probabilities. It makes reference to $Y_t$ without specifying which $t \in M$ is used to define $C_{r,c}$. As a result, after having read the paper 6 times over, I still do not know how to compute $C_{r,c}$.
>
> There is a typo in the equation for $C_{r,c}$: The letter $t$ should be replaced with the letter $c$. We apologize and hope that correcting this typo eliminates any confusion about LossPred.
>
> **Regarding "References to items not yet introduced":**
> >   * Lines 153-154, the phrase "our loss" make it sound like a referrence to the masked language model loss discussed in the previous sub-section, whereas in fact it is referring to Equation (3), a yet-to-be-introduced loss for SylBoost.
>
> We will replace “our loss” with “our loss (Equation 3).”
>
> > * Lines 159-162 give a very vague description of the "similarity matrix" and the "cut algorithm" which can only be known if the reader has already seen the subsequent Section 3.3.
>
> We apologize that the sentence starting on line 159 is written awkwardly. It should instead read “This results in a matrix of pairwise frame distances where element $i,j$ represents the $L_2$ distance between the features at frame $i$ and frame $j$.”
>
> >  * Starting at line 188, Section 4.2 makes repeated references to "mHuBERT". "mHuBERT" appears to be name given to the acoustic tokens in (Hassid et al., 2023) by this paper (line 241). (Hassid et al., 2023) itself does not use this name, so an ordinary reader would not be able to tell what an "mHuBERT" model is when they work through Section 4.2.
>
> Agreed, and we will fix this confusion by replacing “mHuBERT” with “TWIST Tokenizer.” The name mHuBERT was taken from the [github repository](https://github.com/facebookresearch/textlesslib/tree/ba33d669d8284b4f7bfe81e7384e83ab799fe384/textless) published by [Hassid et al](https://proceedings.neurips.cc/paper_files/paper/2023/file/c859b99b5d717c9035e79d43dfd69435-Paper-Conference.pdf).
>
> **Regarding "Confusing terminology":**
> >   * "pretraining": This paper makes a liberal use of the term "pretraining" to the point it's very difficult to tell which is the model being "pretrained". For example,
> >     * Line 113 mentions a "pretrained HuBERT teacher model", then line 119 says "during pretraining, the **student** model ...". The teacher and the student are presumably not trained at the same time, yet the use of "pretraining" in this context make it appear that the contrary is happening.
> >     * Line 225 says "for all pretraining experiments". A reader will have to look really closely to see this means "training of the speech LM", not "pretraining HuBERT, etc".
>
> We will alter Line 225 to say “for all language model pretraining experiments” and in line 119 replace “pretraining” with “training.” We believe that lines 113-114: “We consider the setting of having a pretrained HuBERT teacher model and a HuBERT student model trained to predict the quantized contextualized representations generated by the teacher” make training order clear.
>
> >   * "Agglomeration" vs "SylBoost": This paper appears to use these two terms interchangeably. Agglomerative clustering is apparently also used (line 183). This makes it difficult for the reader to tell when "agglomeration" is mentioned, whether the authors intend to refer to SylBoost or just the clustering.
>
> You are correct that “Agglomeration” and “SylBoost” are used interchangeably, and we will replace these instances of “Agglomeration” with “SylBoost.” We will maintain that we always refer to clustering for discrete tokenization with K-Means as the full phrase “K-Means and Agglomerative Clustering”
>
> **Answering "Is LossPred really necessary?"**
> > * While Table 4 shows that LossPred is a better initialization strategy for SylBoost, would a cheaper initialization strategy (like feature similarity or even random) produce equally good segmentation given more iterations?
>
> In rows 2 and 3 of Table 4, we demonstrate that the second iteration of SylBoost using feature similarity results in worse performance than the first iteration so we do believe that LossPred is necessary.
>
> > * Suppose the answer to the previous question is yes, would it make sense to run SylBoost but using the final activation of HuBERT instead of an intermediate layer (line 154)? The final activation may be better semantic features.
>
> Although the answer to the first question is no, [Pasad et al](https://aclanthology.org/2024.tacl-1.21/) [43] shows that the semantic features from mean-pooling across ground truth syllable boundaries perform best at Layer 9 and not the final layer activations.
>
> If you agree that these minor edits address your major concerns about the paper’s clarity, given that you state that our method and results in this paper would make a good paper for NeurIPS, we kindly ask that you consider raising your score.

---

> > ### Comment · Reviewer_kvPb · 2024-08-07
> >
> > Thank you for taking the time to address my review! I have read your rebuttal and will incorporate the new information in my final feedback.

---

### Official Review · Reviewer_RUrK · 2024-07-13

**Soundness:** 4
**Presentation:** 4
**Contribution:** 4
**Rating:** 8
**Confidence:** 3

**Summary:**

This paper first introduces an algorithm named LossPred that generates syllable-level speech segmentation without any training or supervision. The algorithm works by analyzing the prediction loss of speech tokens under different mask positions.

With the initial boundaries proposed by LossPred, the paper proposes further training a pretrained HuBERT / data2vec2 model by minimizing the sum of squared distances between feature vectors of each token and the average of feature vectors within the corresponding segment. This process is called SylBoost, and it further improves syllabic segmentation performance and efficiency.

Finally, the paper proposes training a Generative Spoken Language Model (GSLM) with the speech tokens obtained from quantized SylBoost units. Compared to existing GSLMs trained on other discrete representations, SylBoost encodes speech into much shorter sequences, significantly boosting training and inference efficiency.

**Strengths:**

1. The proposed speech representation learning and unit discovery algorithms, LossPred and SylBoost, are novel. While the idea of improving computational efficiency through dynamic or fixed-rate downsampling of speech representation is not new, this paper appears to be the first to successfully apply dynamic-rate downsampled representations with a very low sampling rate of 5Hz to Generative Spoken Language Models (GSLMs).
2. The presentation of the paper is of high quality and clarity. The authors report extensive experimental results, which effectively demonstrate that the proposed method outperforms various state-of-the-art (SotA) methods.
3. The topic addressed in this paper is significant, as very low sampling rate speech representations can benefit various tasks, including speech understanding and generation.

**Weaknesses:**

1. As pointed out by the authors, the proposed LossPred and SylBoost methods seem to be restricted to speech representation learning. It might be difficult to apply these methods to music, singing voice, speech with noisy backgrounds.
2. LossPred is slow in evaluating the loss prediction matrix. Each sentence requires about 200 Transformer network evaluations.
3. LossPred is highly heuristic. There seems to be no theoretical guarantee that the HuBERT model combined with LossPred reveals syllabic boundaries instead of revealing only phoneme or word boundaries.

**Questions:**

1. The equation above line 127 is rather confusing. The description of the procedure for computing the loss prediction matrix is confusing in general.
2. In LossPred, $k$, the number of segments, is chosen to be proportional to the sequence length. What happens when a speaker is speaking very fast or very slow? It seems that $k$ should be determined by the number of syllables in the utterance.

**Limitations:**

The authors have adequately addressed the limitations and potential negative societal impact of their work.

---

> ### Author Rebuttal · Authors · 2024-08-05
>
> We sincerely thank the reviewer for their feedback. We hope to clarify any additional concerns they had below:
>
> **Weaknesses**
>
> > 1. As pointed out by the authors, the proposed LossPred and SylBoost methods seem to be restricted to speech representation learning. It might be difficult to apply these methods to music, singing voice, speech with noisy backgrounds.
>
> We entirely agree and hope that future work can tackle these challenges.
>
> > 2. LossPred is slow in evaluating the loss prediction matrix. Each sentence requires about 200 Transformer network evaluations.
>
> This is entirely true. Fortunately, the forward passes can be batched and the first iteration of SylBoost works well with only 10% of LibriSpeech labeled by LossPred.
>
> > 3. LossPred is highly heuristic. There seems to be no theoretical guarantee that the HuBERT model combined with LossPred reveals syllabic boundaries instead of revealing only phoneme or word boundaries.
>
> We agree that there is no guarantee of the types of units that LossPred predicts. We think that an especially interesting case is how the two words “at a” in Figure 1 get mapped to a single cluster by SylBoost. We are hopeful that future work can inspect these phenomena with respect to the statistics of spoken language.
>
> **Questions**
>
> > 1. The equation above line 127 is rather confusing. The description of the procedure for computing the loss prediction matrix is confusing in general.
>
> We sincerely apologize that there is a typo in the referenced equation, and the letter $t$ should be replaced with $c$ in said equation and its description starting on line 128. We acknowledge that LossPred is a detailed algorithm, and fully understand any misunderstanding that could have resulted from this typo.
>
> > In LossPred, $k$, the number of segments, is chosen to be proportional to the sequence length. What happens when a speaker is speaking very fast or very slow? It seems that $k$ should be determined by the number of syllables in the utterance.
>
> We would be interested in a future approach that could attempt to count the number of distinct regions generated by $C$ in LossPred however found that using an empirical $k$ performs well and is compatible with prior cut algorithms such as that used by [Peng et al](https://arxiv.org/pdf/2305.11435).
>
> We thank the reviewer again for their feedback.

---

> > ### Comment · Reviewer_RUrK · 2024-08-12
> >
> > Thank you very much for responding to all of my comments and questions.

---

### Official Review · Reviewer_zmRG · 2024-07-22

**Soundness:** 3
**Presentation:** 3
**Contribution:** 3
**Rating:** 6
**Confidence:** 3

**Summary:**

This paper studies learning low bitrate speech units that preserves semantic information. As presented in the paper, the proposed approaches achieve SoTA performance on tasks like ASR and ZeroSpeech. The proposed approach also shows benefits in terms of compute resources — as claimed by the authors, 30x faster to train, and also benefits in terms of inference and transmission due to low bitrate.

**Strengths:**

Overall, the proposed multistage approach — first using the HuBERT like model to extract syllable-like noisy segmentation, then bootstrapping pseudo-syllabic units iteratively makes sense to me. The proposed approach also shows clear benefits in terms of performance and efficiency.

Good performance: Compared to baseline approaches like SD-HuBERT, the proposed method achieved higher accuracy on syllable boundary detection and clustering, ASR, and also shows better continuation metrics as shown in Table 7 for generative spoken language modeling experiments. All those evaluations all positively demonstrate the strong associations with syllables of the generated speech units, while it does show lier-bitrate compared to the baselines compared in the paper.
The authors also conducted ablation studies to further demonstrate a couple design choices.



Efficiency: As claimed in the paper, the proposed technique is capable of achieving extremely low-bitrate compared to the counterpart speech units, while still being able to achieve good performance in a wide range of tasks, with the efficiency in both training and inference phases.

**Weaknesses:**

Demonstrating efficiency: As efficiency is also one selling point of the paper, it would be great if the authors can demonstrate the training efficiency and low-bitrate benefits in a more comprehensive way, like visualizing the GPU training time vs Performance, and also bitrate vs unit quality for certain tasks.



Limited use cases: The proposed approach focuses on learning semantic units for speech applications. It’s unclear if the proposed methods can be applied to other important non-speech use cases like understanding acoustic environment, and understanding speaker’s identity and emotion.



Understanding Unit Quality: To demonstrate the unit quality for synthesizing the audio and for generation, should the author also compare with other related works (like [1] and [2]) in terms of reconstructing the original signal? Like in [1] (see Table 1), the authors compare the different approaches in terms of reconstruction performance using a couple of metrics like MEL, STFT and ViSQOL score, and also semantic task performance.

[1]: https://arxiv.org/abs/2405.00233
[2]: https://arxiv.org/abs/2306.06546

**Questions:**

See Weaknesses section

**Limitations:**

Not aware of potential negative societal impacts

---

> ### Author Rebuttal · Authors · 2024-08-05
>
> We thank the reviewer for their valuable feedback and for stating the proposed approach shows clear benefits in terms of performance and efficiency. We respond to each of the reviewer’s concerns below.
>
> > Demonstrating efficiency: As efficiency is also one selling point of the paper, it would be great if the authors can demonstrate the training efficiency and low-bitrate benefits in a more comprehensive way, like visualizing the GPU training time vs Performance, and also bitrate vs unit quality for certain tasks.
>
> We demonstrate the GPU training time vs Performance on the semantic tasks of sWUGGY, sBLIMP, and tStoryCloze in Table 3. We demonstrate bitrate vs unit quality for WER and CER measures of quality in Table 2 and further ablate controling for the number of units. We also demonstrate bitrate vs unit quality on sWUGGY and sBLIMP in Table 6, controlling for changes in the number of tokens and for changes in sample rate. To better visualize the role of bitrate in Table 6, we will add a column containing bitrate, however we note that bitrate is directly calculable from the provided unit frequency and number of unit clusters. If you are asking in particular for visual diagrams, such as a GPU-Hour vs sWUGGY accuracy graph, please let us know and we would be more than happy to adapt the results from these tables into a graph in our appendix.
>
> > Limited use cases: The proposed approach focuses on learning semantic units for speech applications. It’s unclear if the proposed methods can be applied to other important non-speech use cases like understanding acoustic environment, and understanding speaker’s identity and emotion.
>
> We fully agree with this limitation and acknowledge it our work. The speech domain is both difficult and important, and so we believe that expanding this paper to evaluate other domains may cause it to lose focus. We believe that focusing in-depth on speech applications throughout our paper provides a strong starting ground for future work to adapt LossPred and SylBoost to other domains, which we eagerly await.
>
> > Understanding Unit Quality: To demonstrate the unit quality for synthesizing the audio and for generation, should the author also compare with other related works (like 1 and 2) in terms of reconstructing the original signal? Like in 1 (see Table 1), the authors compare the different approaches in terms of reconstruction performance using a couple of metrics like MEL, STFT and ViSQOL score, and also semantic task performance.
>
> We believe that evaluating the acoustic quality of our unit reconstruction is not a task best suited for this paper. Like [Hassid et al](https://proceedings.neurips.cc/paper_files/paper/2023/file/c859b99b5d717c9035e79d43dfd69435-Paper-Conference.pdf), we evaluate our units solely on semantic tasks instead of metrics like MEL, STFT, and ViSQOL. Codecs geared toward natural sounding speech generation, such as 1 and 2, are orthogonal to our method and operate at a significantly higher bitrate. Our paper seeks to improve the textual understanding of models, and units like those of 1 and 2 are compatible with downstream vocoding in cascaded networks such as our Interleaved-Vocoder-LM or the fine acoustic modeling stage of [Boros et al](https://arxiv.org/pdf/2209.03143).
>
> We also point out that the WER of the concurrent work (1) referenced is 19.6% at 360bps compared to our 7.0% WER at 81bps (albeit on different datasets), which we hope additionally clarifies how our units cover different problem niches.
>
> If the reviewer believes that the provided response adequately addresses their concerts, we respectfully request that they consider raising their score to reflect that.

---

### Official Review · Reviewer_zj85 · 2024-07-23

**Soundness:** 2
**Presentation:** 2
**Contribution:** 2
**Rating:** 3
**Confidence:** 3

**Summary:**

This paper proposes an approach for extracting syllable-like units from speech SSL models for use in a transformer-based language model. The motivation is that, compared to baseline acoustic units, which tend to mimic phonetic units in their time resolution, syllable-like units have lower time resolution, which makes them easier to model using techniques from the language domain. The authors propose an adaptation of the SD-HUBERT approach to extract units that can be used in Generative Spoken Language Modeling.

**Strengths:**

The authors identify an important limitation of why using language modeling techniques is a challenge in the speech domain, and their proposed approach seeks to address the limitation.

**Weaknesses:**

Overall, I found the submission difficult to follow. Please see my additional comments below.

line 2 -> Transformers do not require the inputs to be tokenized. The tokenization step is performed so that we can use language modeling techniques in speech.

line 18 -> Generally speaking, there is no requirement for the SSL representations to be powerful or abstract.

Line 20 -> I don't see how the example of young children motivates your SSL description from the previous sentence; the transition is incoherent.

line 22 -> What does performant mean in this case? What is the connection between composing highly realistic text and the ability of a model to provide features for a downstream task? You seem to conflate the two goals, even though they are not necessarily the same.

line 23 -> The statement on this line is not clear. Several successful speech language model methods were introduced in the literature, what about previous approaches that make them fail? Please consider clarifying.

line 31 -> The temporal resolution impacts the LM part of the problem. Why is it important if we want to extract features for a downstream task?

line 37 -> What does "syllable-like" mean in this case? Can you elaborate on the time resolution it represents? Why is it important to start with a "syllable-like" unit? What makes it suitable for GSLM? What challenges from prior work are you addressing when using "syllable-like" units?

Line 38 -> I would refrain from using words like "breakthrough" and instead let the reader decide if the improvement is indeed a "breakthrough."

line 48 -> I disagree with labeling your method as "train-free" since it relies on a pre-trained HuBERT model.

line 51 -> The distinction between the first and second contributions needs to be clarified. If the boundaries from the first contributions are not good on their own, then why mention them as a contribution?

Line 102 -> It is not clear how/where you do the masking. Do you do it on the raw input, mel-spectrogram, or the extracted features?

line 113 -> Shouldn't the approach be "train-free"? Why do we have a student/teacher model that we are training?

line 147 -> The authors must refine the motivation for why syllabic units are useful for this application. Why not use word units instead?

line 189 -> Superior compared to what?

line 198 -> I suggest leaving any experimental details to the experiments sections.

Table 1 -> Can you try any non-neural baselines for boundary detection? What would the performance be if we used heuristics based on energy, zero-crossing rate, or changes in Prosody to get rough boundaries?

Table 1 -> What makes Data2Vec2 better than HuBERT for extracting boundaries?

Table 1 -> What happens if you apply SylBoost to Feat-Sim?

Table 1 -> Please describe the metrics and abbreviations in the captions.

Table 2 -> What does the underline represent?

line 221 -> Implement what exactly? Please re-write the sentence.

line 237 -> typo

Table 3 -> What does the underline represent?

*Estimated.-> What does estimated mean? If prior work does not explicitly give this information, then it is better to leave it out.

line 263 -> What is the R score?

line 281 -> Please present the tables in the order they are referenced in the text; you currently jump from Table 1 to Table 4 and then go back to Tables 2 and 3.

line 340 -> Communication is not the last name of the first author from [16]

**Questions:**

How does the performance on the syllable boundary detection task relate to the unit re-synthesis and language modeling performance? In other words, is having well-defined boundaries necessary for the speech language modeling task?

It is difficult to disentangle the effect of the number of units and temporal resolution on the overall performance in Tables 2 and 3. Despite having lower resolutions, the results from the proposed methods are not much better than baseline methods (e.g., AudioLM and TWIST). Can the authors elaborate on this?

What happens if we directly use LossPred units in the LM framework? In other words, how important is the boundary refinement stage for the language modeling task?

Keeping everything fixed, how does the number of units impact the re-synthesis and language modeling results?

How does the masking probability and span used when training the HuBERT model impact the quality of the discovered boundaries, and what impact does it have on your approach?

**Limitations:**

Can the authors comment on the trade-off between resolution and ease of modeling (and quality)? What do we lose/gain using syllable-like speech units in a language modeling paradigm?

---

> ### Author Rebuttal · Authors · 2024-08-05
>
> Thank you for the review. Below, we will respond to several questions you raised about our work.
>
> > line 189 -> Superior compared to what?
>
> This is answered in lines 189-190: “Superior performance in high difficulty settings **compared to other Neural Codec Lanaugae Models like VALL-E [54]**”
>
> > Table 1 -> What happens if you apply SylBoost to Feat-Sim?
>
> These results are the focus of Table 4, where we evaluate SylBoost from both FeatSim and LossPred and find that initializing with FeatSim boundaries results in SylBoost converging to lower-quality boundaries.
>
> **Questions**
> > How does the performance on the syllable boundary detection task relate to the unit re-synthesis and language modeling performance? In other words, is having well-defined boundaries necessary for the speech language modeling task?
>
> Higher quality boundaries are essential for low error rates, a comparison which we draw from the WER of SD-HuBERT and SylBoost+HuBERT units in Table 2. Having exact syllable boundaries is not necessarily essential for the language modeling task after word error rates are low, however, as seen through evaluating on a wide range of unit rates (5.0-8.33 Hz) in Tables 2, 6, and 7.
>
> > It is difficult to disentangle the effect of the number of units and temporal resolution on the overall performance in Tables 2 and 3. Despite having lower resolutions, the results from the proposed methods are not much better than baseline methods (e.g., AudioLM and TWIST). Can the authors elaborate on this?
>
> We respectfully disagree with the framing of this question. First, we heavily outperform TWIST-CI-90M and TWIST-300M with our proposed models, where we can afford to match training compute. We also beat AudioLM in sWUGGY with less than 10% of the pretraining compute, and are the first work to come within 1% of AudioLM on sBLIMP. SyllableLM only starts losing to TWIST models when they reach 7B parameters, 3x the data, and 20x the pretraining GPU compute hours.
>
> Second, we disagree with “Despite having lower resolutions.” Lower resolutions provide a significant speedup as generation and training run on up to 5x fewer tokens so even equivalent performance would be significant. Using fewer tokens is a challenge, and our work is the first to reach below 19.5Hz.
>
> > What happens if we directly use LossPred units in the LM framework? In other words, how important is the boundary refinement stage for the language modeling task?
>
> LossPred only provides boundaries and not units, as it is calculated only using a model loss instead of a cut-algorithm on model features. Even if we used the LossPred boundaries for HuBERT pooling, LossPred is too expensive to extract across an entire dataset, which is one reason we created SylBoost.
>
> > Keeping everything fixed, how does the number of units impact the re-synthesis and language modeling results?
>
> Resynthesis results, keeping everything else fixed, are in the “+Increase #Units” row of Table 2. The language modeling results are in “Table 6: Holding number of units and unit rate constant.”
>
> > How does the masking probability and span used when training the HuBERT model impact the quality of the discovered boundaries, and what impact does it have on your approach?
>
> We do not have the compute to pretrain a HuBERT model with the requested modifications and we are not aware of any public checkpoints that would facilitate this, so unfortunately we cannot answer this question.

---

> > ### Comment · Reviewer_zj85 · 2024-08-12
> >
> > Thank you for the clarifying comments. I will incorporate this information in my final feedback.

---

### Decision · Program_Chairs · 2024-09-25

**Decision:**

Reject

**Comment:**

**Summary**

The paper proposes a new tokenization, syllable-like units, of speech on top of the SSL representations for decoder-only models training in generative spoken language modeling. It uses iterative process to generate the segmentation and then apply clustering on top of the mean pooled features. Authors claim that this reduces the bitrate significantly which makes the language modeling more efficient — the model achieves similar performance as SOTA models while being significantly more efficient.

**Arguments taken for the decision**
- **Reviewer kvPb, RUrK, zmRG** pointed out that the idea on the process to produce syllable-like segmentation of speech in an unsupervised manner is new and the paper brings strong contribution in terms of a lower inference computational cost.
- **Reviewer kvPb** thinks that the design of experiments is sound. In addition **Reviewers RUrK, zmRG** pointed out that Authors report extensive experimental results.
- **Reviewers RUrK, zmRG** pointed out that LossPred is slow in evaluating the loss prediction matrix, may not be applicable to music, speech with background, and is kind of really heuristic. **In the rebuttal** Authors agreed with these points.
- **Reviewer zj85** found it is hard to follow submission and pointed to a lot of places which were unclear. Authors clarified some of the points **in the rebuttal**.
- **Reviewer kvPb** believes that paper needs substantial revision and thus could not be accepted at the current state. During **rebuttal** Authors provided clarifications and the changes they can make to improve readability.
- **Reviewer zmRG** pointed out that clear demonstration of the method efficiency is needed and also comparison with other unit quality reconstruction baselines. **In the rebuttal** Authors pointed out that efficiency is given in Tables, while the acoustic quality of proposed units reconstruction is not a task best suited for the paper.
	- AC comment: I disagree with Authors on the latter point, as otherwise it means that you need to have in addition very strong acoustic tokens to avoid losing information you lose with lowering bitrate even for semantic HuBERT tokens. Moreover you should be doing more heavy work on the detokenization part then, which can be inefficient.
- **After rebuttal discussion:** both Reviewers kvPb and zj85 still point out that substantial changes are needed for the paper readability  as otherwise it is hard to follow paper despite Authors' points.

**Recommendation**

The reviews for the paper are mixed, and during AC-reviewers discussion period **Reviewers kvPb and zj85** are mainly concerned with the paper writing style and substantial changes needed to improve the readability and clarity of the paper. In addition to reviewers I also did a pass over the paper and I agree with reviewers: in many places it is hard to follow, a lot of things are mixed up and language in the paper should be improved. Based on that **I recommend rejection of the paper in its current form**. I agree with reviewers that the paper has valuable contributions though articulation of results is not clear right now.

**Recommendation to Authors**

I encourage Authors carefully rework the text with clear definitions, formulas, summaries and ideas formulation, having explanation of all main pieces clearly assuming less background knowledge from the readers (e.g. formulating things exactly rather than just citing - e.g. how you perform even unit to speech mapping), and then resubmit to the next venue. Also I encourage Authors to look into dependence on the acoustic tokens extraction and investigate deeper if the proposed method applicable across different speech tokenization or it is specific to HuBERT ones only. This will strengthen the paper significantly.